# Safety, Effect and Feasibility of Percutaneous SI-Screw with and without Augmentation—A 15-Year Retrospective Analysis on over 640 Screws

**DOI:** 10.3390/jcm9082660

**Published:** 2020-08-17

**Authors:** René Hartensuer, Moritz F. Lodde, Jonas Keller, Maria Eveslage, Josef Stolberg-Stolberg, Oliver Riesenbeck, Michael J. Raschke

**Affiliations:** 1Department of Trauma-, Hand- and Reconstructive Surgery, University of Muenster, 48149 Münster, Germany; moritz.lodde@ukmuenster.de (M.F.L.); j_kell13@uni-muenster.de (J.K.); Josef.Stolberg-Stolberg@ukmuenster.de (J.S.-S.); oliver.riesenbeck@ukmuenster.de (O.R.); michael.raschke@ukmuenster.de (M.J.R.); 2Institute of Biostatistics and Clinical Research, University of Muenster, 48149 Münster, Germany; maria.eveslage@ukmuenster.de

**Keywords:** sacroiliac-screw, SI-screw, percutaneous operation, augmentation, pelvic ring, fragility fracture, 3-D navigation

## Abstract

Background: Minimally invasive sacroiliac-screw (SI-screw) fixation of the pelvis is used in energy trauma (Arbeitsgemeinschaft für Osteosynthesefragen (AO) classified) and fragility fractures (Fragility Fracture of the Pelvis (FFP) classified). However, available clinical data are based on small case series and biomechanical data seem to be contradictory. Methods: The present single center retrospective cohort study investigated percutaneous SI-screw fixation and augmentation over 15 years. Groups were compared concerning the general epidemiological data, mobilization, complication rates, duration of stay, and safety of SI-screw insertion. Multivariable analyses were performed using logistic regression. Results: Between 2005 and March 2020, 448 patients with 642 inserted SI-screws were identified. Iatrogenic neurological impairment was documented in 2.47% and correlated with screw misplacement. There was an increased complication risk in patients with AO type C over patients with AO type B and in FFP II over FFP III/IV patients. Cement-augmented FFP patients showed a 25% reduced stay in hospital and a reduced complication risk. Cement-associated complications were seen in 22% without correlation to neurologic impairment. Conclusions: The present study confirms the safety and usability of percutaneous SI-screw fixation, despite specific risks. Cement augmentation seems to reduce the complication risk in FFP patients and shorten hospital stay for some reasons, without increased specific complications or correlated neurological impairment.

## 1. Introduction

Percutaneous sacroiliac-screw (SI-screw) fixation is considered to be the “gold standard” for the treatment of posterior pelvic ring fractures since it was described in 1973 [1]. However, there is still a considerable risk of iatrogenic injury to the lumbo-sacral nerve roots, the superior gluteal artery and the iliac vessels [2]. These complications can be caused by K-wire or screw mal-positioning during insertion. The rate of mal-positioning is still reported to be approximately 5% [3,4]. Nevertheless, the advantages over open techniques are obvious and some authors consider percutaneous SI-screw fixation to be the only minimally invasive technique to stabilize the posterior pelvic ring [5]. Traditionally, pelvic injuries are considered to be associated with an immense traumatic impact, often associated with other severe injuries. An increasing number of cases, however, resulted from low energy trauma associated with osteoporosis in the geriatric population. Different classifications are available to rate these injuries. One common classification was introduced by Tile [6] and is included into the AO classification systematics [7]. To prevent complications, early mobilization is one reason for surgery. Despite the increasing usage of the minimally invasive SI-screw technique, knowledge is still limited. Recent reviews are based on small case series, biomechanical investigations, and technical reports, so far [8,9].

The rising numbers of fractures to the pelvis occurs in the aging population. Here is an increasing incidence of fractures to the pelvis as a result of low impact and associated with an advanced frailty status [10]. It seems to be crucial to distinguish between these two entities. The so-called fragility fractures are described to differ in terms of morphology and injury mechanisms from high-energy pelvic injuries. The grade of instability in these fractures may increase over time [11]. In 2013 Rommens and Hofmann proposed a new classification for the fragility fractures of the pelvis (FFP) [12]. The FFP classification is reported to allow moderate intra- and inter-rater reliability [13]. A disproportionately high in-hospital mortality rate with these, in supposedly less severely injured patients, is reported [14]. Pain reduction and rapid return to mobility are the main treatment goals in these patients [15]. Eckardt et al. described a good functional outcome after percutaneous screw stabilization in patients suffering fragility fractures of the pelvis [16]. They investigated 50 patients with a mean age of 79 years. They described a 1-year mortality of 10% and an additional loss of independency in 13%. Furthermore, they reported on one intraoperative screw misplacement, one severe intraoperative bleeding and one post-operatively detected misplaced screw with ischial pain. Moreover, implant loosening in over 30% with 18% of patients requiring revision surgery is reported [16]. Despite the wide usage of percutaneous SI-screw fixation for the treatment of pelvic ring fractures, this technique remains insufficient for the anchorage of the screws inside the sacrum in cases of severe osteoporosis. To gain increased anchoring, Tjardes et al., as well as Müller and Fuchtmeier, suggested cement augmentation [17,18], which was described by Wähnert et al. using cannulated perforated SI-screws [19]. To data, few data are available on this topic. A recent systematic review by König et al. found eleven studies. Out of these, five were case series and the other six were biomechanical cadaveric studies [9]. They concluded that, based on few clinical case series with relatively low numbers of patients and available biomechanical data, the augmentation of SI-screws was safe and effective. However, the biomechanical results on the effect of augmentation are contradictory. Some authors found no difference between the augmentation of sacroiliac-screw and conventional sacroiliac-screw fixation [20,21,22]. Others report on improved biomechanical stability due to cement augmentation [23,24,25]. Clinical data again are only available from small case series. The number of patients ranges from 8 to 34 [19,26,27,28,29]. König et al. concluded in their review that there were no larger case series, prospective data or randomized trials available [9]. To our knowledge, these important contributions to estimate the clinical impact are still missing.

We were able to retrospectively investigate percutaneous SI-screw fixation and augmentation in a large number of patients treated over the last 15 years. These single center retrospective observational data should provide an important contribution to the remaining uncertainty about risk, safety and immediate effects of SI-screw fixation, with and without augmentation.

## 2. Experimental Section

This is a retrospective, single-center cohort study. From January 2005 to March 2020, patients were identified by operation and procedure coding (OPS) (5-790.0d, 5-79a.0e, 5-79b.0e, 5-83b.20, 5-83b.21) using the hospital database. Individual case review was performed for each case, including digitally available imaging. Retrospective data acquisition included age, sex, numbers of SI-screws, cement augmentation, classification of injury, Patient Clinical Complexity Level (PCCL) [30], American Society of Anesthesiologists (ASA) score, Charlson Comorbidity Index (CCI) [31], body mass index (BMI), duration of in-hospital stay, major and minor complications, screw-associated complications, cement=associated complications, mobility at time of discharge, usage of intraoperative 3D scan, usage of intraoperative navigation and postoperative peripheral neurological complications. All documented complications were recorded and rated to be minor or major complications. An overview is given in Table 1. All cases were re-classified using available imaging or the description of morphology. Descriptive analysis was performed including all patients providing the required data. The number of included patients for distinct analysis was provided where appropriate. An overview is given in the result section (Figure 1). Pelvic ring fractures caused by high-energy trauma were classified using the Arbeitsgemeinschaft für Osteosynthesefragen (AO) Classification [7]. All fractures caused by minor trauma were classified using the FFP classification [12]. By doing so, we were able to distinguish between high- and low-energy trauma in our data set. Screw-associated complications distinguished between the non-perforation of cortical bone, less than 3.6 mm (mild) and more or equal than 3.6 mm (severe), regardless of potential clinical significance. This grading was chosen in context with the used 7.3 mm and 7.5 mm screws. Cement-associated complications were defined to include all cement extrusions outside the bony sacrum, regardless of potential clinical significance. Clinically obvious problems, like postoperative peripheral neurological impairment, were recorded and reported separately.

Statistical analyses were performed using the SAS^®^ software version 9.4 for Windows (SAS Institute, Cary, NC, USA). Data are described using the median and interquartile range (Q1–Q3) or the absolute and relative frequencies. Groups were compared concerning the complication rates (i.e., proportion of patients with at least one major/minor complication) using Fisher’s exact test. Multivariable analyses were performed using logistic regression. Results are reported as odds ratios (OR) and the corresponding 95% confidence intervals. Groups were compared concerning the ordinal and continuous variables using the Mann–Whitney *U* test. Linear regression was used to analyze the duration of hospital stay, which was log-transformed because of its skewed distribution. Therefore, the results are reported in terms of ratios concerning the geometric mean and corresponding 95% confidence intervals. Missing data are indicated for each analysis in the results section.

The study was approved from the local ethical committee (Ref.#2020-409-f-S).

## 3. Results

Between 2005 and March 2020, 448 patients with 642 inserted SI-screws were identified. The median age was 57.93 years (Q1:40.97; Q3:77.16). Overall sex distribution f/m was 249/199 (56%/44%). Median age of males was 49.72 years (Q1:31.07; Q3:65.84); median age of females was 70.90 years (Q1:48.83; Q3:81.59). 

One hundred and twenty-four patients (27.68%) were classified as AO B, 176 patients (39.29%) as AO C, 117 patients (26.12%) as FFP II, nine patients (2.01%) as FFP III and 22 patients (4.91%) as FFP IV. There was an obvious difference in age and sex between the trauma categories. The median age in AO B patients was 49.35 years (Q1 31.04; Q3 64.71). In AO C patients the median age was 46.49 years (Q1 28.50; Q3 60.98). The median age in FFP II patients was 80.16 years (Q1 73.09; Q3 85.36), in FFP III patients 71.61 years (Q1 65.96; Q3 81.88) and in FFP IV patients 76.31 years (Q1 69.90; Q3 80.81), respectively. 

Sex distribution in both AO groups (*n* = 300) was f/m 39.67%/60.33% and in all FFP groups (*n* = 148) f/m 87.84%;12.16%, respectively. The different distribution of age separated by sex is presented in Figure 2.

BMI calculation was possible in 362 patients. Median BMI was 24.49 (Q1: 22.28; Q3: 27.68) ranging from 15.76 (min) to 50.70 (max). No differences between males and females were found.

A single screw was inserted in 282 (63.09%) patients, two screws in 145 (32.44%) patients and more than two screws in 20 (4.47%) patients. Intraoperative 3D imaging was documented in 69%. Consecutively, in 31%, intraoperative conventional 2D fluoroscopic imaging was documented. Out of the 305 cases in which 3D imaging was documented, one single scan was performed in 246 cases (80.66%), two scans were documented in 47 (15.40%) cases and three scans were documented in 12 cases (0.04%). Navigation was used in 141 cases (31.69%). Intraoperative 3D scans performed for image acquisition to perform 3D navigation was not counted for the above presented interoperative 3D imaging. Out of the navigated cases, five cases of 2D navigation were documented. All the others were 3D navigated.

Overall documented PCCL coding was distributed with two peaks. PCCL 0 was found in 101 cases (22.54%), whereas PCCL 1 and PCCL 2 were documented only in 22 (4.91%) and 44 (9.82%) cases, respectively. Another peak was seen in PCCL 3 and PCCL 4 with 117 (26.12%) and 100 (22.32%), respectively. PCCL 5 was found in 51 cases (11.38%) and PCCL 6 in 13 (2.90%). The majority of cases were rated ASA 2 and 3 (ASA 1: 18.66%; ASA 2: 45.68%; ASA 3: 33.70%; ASA 4: 1.95%), whereas the majority of cases presented a CCI of 0 (CCI 0: 64.29%; CCI 1:15.18%; CCI 2: 10.04%; CCI 3: 5.36%; CCI 4: 2.01%; CCI 5: 1.79%; CCI 6: 0.89%; CCI 7: 0.45%). The median age in the cases with CCI 0 was 48.13 years (Q1: 29.31; Q3:63.90) whereas the median age in CCI 1–7 was higher than 70 years. In all patients, the major and minor complications were seen more frequently in males. Major complication rate (patients with at least one complication presented in Table 1) in males (17.09%) was twice as much compared to that in females (8.84%) (*p* = 0.0097). Moreover, minor complications occurred more often in males (29.65%) than in females (20.56%) (*p* = 0.0278). Multivariable regression analysis also shows a reduced risk of major complications in females compared to males (OR 0.538), an increased risk in AO C compared to AO B (OR 3.516) with a comparable risk between FFP II and AO B (OR 0.819). Furthermore, we found a considerable increased risk of FFP III/IV over AO B (OR 4.137) and even increased compared to AO C (OR 1.176). Odds ratios with 95% confidence limits for major complications are illustrated and presented in Figure 3a. The analysis for minor complications revealed comparable results illustrated and presented in Figure 3b. Since FFP group was predominant in augmented cases, a separate analysis on this group will be provided separately to facilitate comparability.

### 3.1. Safety of SI-Screw Insertion

Out of the 642 inserted SI-screws in 448 patients, the review of correct placement was possible for 422 patients (604 screws). Correct screw placement was accounted for in 385 patients (91.23%), whereas minor screw displacement was found in 19 patients (4.5%) and major displacement was seen in 13 patients (3.08%). In five cases, revision surgery was performed (1.18%). Iatrogenic neurological impairment after surgery was documented in 11/446 patients (2.47%). In eight cases, neurological deficit was present at the time of discharge, and in three cases, temporary impairment with recovery at time of discharge was documented. Despite the screw misplacement, 5/37 (13.51%) of the consecutive neurological deficits were found in this cohort. However, neurologic complications were correlated with screw misplacement (*p* = 0.0001). There was no difference concerning the observed rate of misplaced screws between the conventional placement (8.05%) and the 3D navigated screw insertion (9.24%). In this context, no reduction of screw-related complications using 3D navigation was seen (*p* = 0.3076) in our collective. The median BMI in patients with screw misplacement was 26.09 mg/m^2^ (Q1 23.37; Q3 29.22) compared to a median BMI of 24.49 mg/m^2^ (Q1 22.22; Q3 27.44) in patients without screw-related complications (*p* = 0.0938).

### 3.2. Analysis of AO-Classified Patients

Median duration of hospital stay in AO B patients was 15 days (Q1 8; Q3 24.5) and obviously shorter compared to 21 days (Q1 13; Q3 34) in the AO C patients. PCCL 0 and 1 was seen predominantly in the AO group. In the AO B group (*n* = 124), mobilization until discharge was performed using Crutches in 69.35%. Rollator or walking frame was necessary in 9.68%, a wheelchair in 11.29% whereas no adequate mobilization until discharge was documented in 8.87%. No information about mobilization was found in 0.81%. In the AO C group (*n* = 176), the rate of cases that were able to use crutches was obviously lower (45.45%). In this group, 26.70% required a wheelchair at the time of discharge, whereas 8.52% were mobilized using a rollator or walking frame. In 16.47%, no adequate mobilization was possible. In the remaining 2.84%, no information about mobilization was documented. As already shown, there was an obviously increased risk for developing a major or minor complication in AO C patients compared to the AO B patients presented in Figure 3.

### 3.3. Analysis of FFP-Classified Patients

In FFP II, the median duration of stay was 8 days (Q1 5; Q3 12), in FFP III 7 days (Q1 7; Q3 8) and in FFP IV 12 days (Q1 10; Q3 18). Overall, 52.03 % of FFP patients were mobilized using a rollator or walking frame. In addition, 29.73% were able to use crutches until discharge. In 7.43%, mobilizing was only possible using a wheelchair, and in 8.79%, mobilization was not successful until discharge. Data on mobilization were missing in the remaining 2.03%. There was no obvious coherence between mobilization and age except for bedridden cases. Median age in this population was about 5 to 10 years older. Details are provided in Table 2. There was no obvious influence of PCCL on the duration of stay except very high PCCL scores 5–6. Details are presented in Table 3.

Major complications in FFP II group were found in 7/117 (5.98%) compared to 2/9 (22.22%) in FFP III and 5/22 (22.73%) in FFP IV. We found 14 major complications in 148 patients in these groups. Moreover, minor complications occurred less in FFP II 20/117 (17.09%) compared to 4/9 (44.44%) in the FFP III and 9/22 (40.91%) in the FFP IV group. Due to similar complication rates in FFP III and FFP IV with relatively low numbers, FFP III and IV were merged for some analysis. In this case, the group is presented as FFP III/IV. There was an obvious difference between the FFP II and the FFP III/IV group, in the occurrence of at least one minor or major complication. Complication rate in the FFP II group was 27/117 (23.08%) compared to 17/31 (54.84%) in the FFP III/IV group. Multivariable regression analysis revealed a reduced risk for major complications with augmentation vs. none (OR 0.598), and a slightly reduced risk for females over males, also in the isolated analysis of FFP groups (OR 0.974). Furthermore, there was an increased risk in FFP III/IV compared to FFP II (OR 4.679), illustrated in Figure 4a. Comparable effects could be detected for minor complications. Further information is illustrated and presented in Figure 4b.

### 3.4. Effect and Safety of SI-Screw Augmentation

Augmentation was performed in 118 patients (26.34%). Out of these, 49 cases (41.53%) were treated with unilateral and 69 (58.47%) with bilateral augmentation. Augmentation was mainly performed in FFP-classified fractures (66.89% augmented vs. 6.33% in AO, *p* <0.0001). For further investigation regarding augmentation, only FFP-classified cases would be included. No statistical differences in BMI between the augmented (median: 24.01, Q1:21.41; Q3:26.57) and non-augmented patients (median: 24.69, Q1: 22.49; Q3: 27.76) were found.

FFP patients with augmented SI-screws showed a higher ASA scoring than the non-augmented (47.3% ASA 3–4 in patients with augmented SI-screws vs. 30.8% in patients without augmentation, *p* = 0.0391). Moreover, CCI was higher in the augmented group compared to the non-augmented (20.8% CCI >3 vs. 10.3%, *p* = 0.150).

Univariate regression revealed a 25% (ratio augmented vs. non-augmented 0.744, 95%CI 0.585–0.945, *p* = 0.0157) reduced time from surgery to discharge (unilateral: median 8 days; bilateral median 7 days; non-augmented median 11 days) in the augmented cases. There were no statistical differences between the unilateral and bi-lateral augmentation (Figure 5).

Mobilization of the FFP patients after surgery was possible in 91.03%. Mobilization in dependence of augmentation showed a higher rate of cases mobilized with a rollator or walking frame, whereas in the non-augmented group, an obvious higher rate of mobilization with crutches was detectable (*p* = 0.0736). Details are provided in Table 4.

There was no statistical difference in the occurrence of major complications between the augmented and non-augmented FFP cases (8/99 vs. 6/49 patients, *p* = 0.5513). Moreover, minor complications were not different with and without augmentation (20/99 vs. 13/49 patients, *p* = 0.407). In FFP patients, neither a difference was seen between female and male for major complications (2/18 m vs. 12/130 f, *p* = 0.680) nor for minor complications (6/18 m vs. 27/130 f, *p* = 0.236). Interestingly, major and minor complications occurred less in FFP II (major: 7/117, 5.98%; minor: 20/117, 17.09%) compared to FFP III/IV (major: 7/31, 22.58%; minor: 13/31, 41.94%). This difference was statistically noticeable for major complications (*p* = 0.0108) and minor complications (*p* = 0.006).

Univariable regression revealed a reduced risk of major complications in augmented FFP cases for bilateral augmentation (OR 0.850) and unilateral augmentation (OR 0.224) compared with no augmentation. A similar effect was explored concerning the minor complications for bilateral augmentation (OR 0.746) and unilateral augmentation (OR 0.615), shown in Figure 6.

Cement-associated complications were seen in 26/118 cases (22%). Out of these, spinal extravasation was detected in 3/118 cases (2.5%) and foraminal extravasation was seen in 2/118 cases (1.7%). All other cement extrusions were without correlations to neurologic structures and no cardio-vascular relevance was documented. Neurologic complications were correlated with documented screw misplacement (Section 3.1.), but not with cement complications (*p* = 0.3936).

## 4. Discussion

Percutaneous SI-screw insertion in our department was performed over the last 15 years. In consequence, we were able to report on 448 patients with 642 inserted SI-screws. To our knowledge, this is the largest clinical study on this topic, so far. We have not investigated any alternative treatment options. For that reason, no comparative conclusion can be provided in this study.

A retrospective, single-center cohort study on 102 traumatic patients was reported by Pishnamaz et al. [33]. Misplacement in conventional placed screws is reported to be between 8.8–12.4% [4,33,34] in conventionally techniques and 3.6–4.9 % using 3D navigation [4,35]. We found correct screw placement in 91.23%, whereas minor screw displacement (<3.6 mm) occurred in 4.5% and major displacement was seen in 3.08%. The neurological sequel of screw mal-positioning was reported to be up to 18% [2]. Our results confirm this by 13.51% of consecutive neurological deficits. Furthermore, peripheral neurologic complications were correlated with screw misplacement (*p* = 0.0001).

The rate of mal-positioning is reported to be reduced in 3D navigated techniques [34,36]. We cannot confirm this from our results and found no statistical differences between the conventional SI-screw insertion and 3D navigation in this context. This is in accordance with a report from the German Pelvic Trauma Registry [4]. However, the investigated time period potentially included some surgical and technical evolutions. For that reason, it is difficult to generally reject potential advantages of 3D navigation based on our results. However, our data contribute to the general judgement that percutaneous SI-screw osteosynthesis is considered to be safe in general, but technically demanding with specific risks [2,36,37].

Most reports on screw misplacement and safety report on traumatic pelvic fractures [2,4,33,34,35,36]. The present study does not differentiate between traumatic and fragility fractures in this specific context. However, the mechanisms of injuries and its pathology are considered to be different [38]. For that reason, we used a differentiated method to evaluate and report our results. To avoid confusion, “traumatic fractures”, resulting from high energy, were classified using the AO classification [6,7] and fragility fractures were classified using the FFP classification [12]. Doing so, we were able to report on both entities separately, but also together where appropriate. In addition to reporting on obvious screw-related complications, multivariable regression analysis revealed that women may have a reduced risk compared to men in developing complications. These findings contradict review-based reports on equal odds for complications in spinal surgery [39]. We also found an increased complication risk and a longer stay in hospital in patients with AO C fractures compared to AO B fractures. These findings were expected and are consistent with current literature [40]. The distribution between AO B type and AO C type may be different to other epidemiologic reports [41]. This effect is potentially explained by the fact that only patients who received SI screw fixation were included in this study. We also revealed more complications and increased hospitalization time in FFP III/IV compared to FFP II. This is consistent with the anticipated increase in severity and instability within the classification [11,12]. Therefore, the usage of both classifications in our study are consistent. We found more complications and a longer stay in hospital in AO C patients compared to AO B. Interestingly, FFP II fractures showed a risk for complications comparable with AO B patients, whereas FFP III and IV revealed an increased risk for complications, even increased over AO C. Taking into account that there is a possibility of progression of fracture severity reported [42,43], further information seems to be mandatory which patients suffering FFP I or II fractures will heal conservatively, and which preferably should be offered early in the surgery to prevent progression. This is probably one of the most important questions in this field to be answered in the near future. 

One additional option to potentially increase screw anchorage in reduced bone quality is cement augmentation. Höch et al. investigated in-screw augmentation in a prospective observational study of 34 patients and reported this technique to be effective and safe [27]. However, a recent review showed that there was little evidence in this field [9]. They reported that the confirmation of safety and effectiveness was based on only few clinical case series with relatively low numbers of patients. They concluded that based on available literature, augmented SI-screw techniques need to be considered to be experimental with unclear clinical benefit [9]. Our results on a relatively large number on patients may contribute to fill this gap of knowledge. No increase in specific complications was seen after the augmentation compared with non-augmented comparable patients. A reduced risk for general complications in the augmented group was seen, instead. Cement-associated complications were seen in 5.82%. This seems to be relatively low in comparison with recently published leakage rates of 14.7% in kyphoplasty [44]. Fragility fractures of the pelvis are reported to be associated with intense and immobilizing pain [11]. This immobilization may cause additional complications. Van Dijk et al. reported a complication rate of 20.2% after the fracture of the pubic ramus [45]. To prevent prolonged immobilization, surgery might facilitate mobilization. We found that immediate mobilization was possible in more than 90% after surgery. Augmented patients were predominately mobilized with a rollator or walking frame, whereas in the non-augmented group, a higher rate of mobilization with crutches was observed. Even if there was no statistically evident effect on the kind of mobilization, univariate regression revealed a 25% reduced time from surgery to discharge in the augmented group. Furthermore, multivariable regression revealed a potentially reduced risk for both minor and major complications, whereas no increased screw-related complication rates were seen between the augmented and non-augmented collective. Due to only 14 major complications in FFP groups, the statistical conclusiveness of multivariable analysis for major complications in these groups (shown in Figure 3) may be limited. However, the results are consistent with other findings presented in this study. Even in consideration of a selection bias in the surgeon’s decision for augmentation, it is conjecturable that this was made in potentially lower bone quality and inferior patient conditions, which is supported by higher ASA and CCI scoring in the augmented group. This would even enhance the shown effect. To the best of our knowledge, this effect was not reported previously and would complement the shown safety with a potential benefit. The reason for this effect remained unclear. Osterhoff described that in a biomechanical study, the mode and dynamics of failure changed due to cement augmentation. However, they found no advantages in terms of screw motion or cycles to failure [21]. This is consistent with other biomechanical studies in which cement augmentation does not prevent from screw loosening and implant failure [22]. Other factors like the stabilization of the anterior pelvic ring potentially influence this mid- and long-term outcome, with a potentially additive effect. However, this was not investigated in this study.

To date, the benefit of cement augmentation in SI-screw osteosynthesis is considered to be unclear and the usage therefore to be experimental [9]. Our data conclude the safe and beneficial usage of cement augmentation of SI-screws in selected patients.

## 5. Conclusions

Percutaneous SI-screw osteosynthesis is confirmed to be safe for traumatic pelvic ring fractures and fragility fractures of the pelvis in general. However, this technique remains technically demanding with specific risks. The potential neurological sequel of screw mal-positioning reported previously was confirmed. Cement augmentation was shown to be safe and effective. We have not found increased specific complications or correlated neurological impairment in this context. Augmentation seems to reduce the risk of general complications in FFP patients. Furthermore, augmented FFP patients have a shorter hospital stay. Based on our results, the augmentation of percutaneous SI-screw osteosynthesis cannot be considered to be experimental with uncertain benefit anymore.

## Figures and Tables

**Figure 1 jcm-09-02660-f001:**
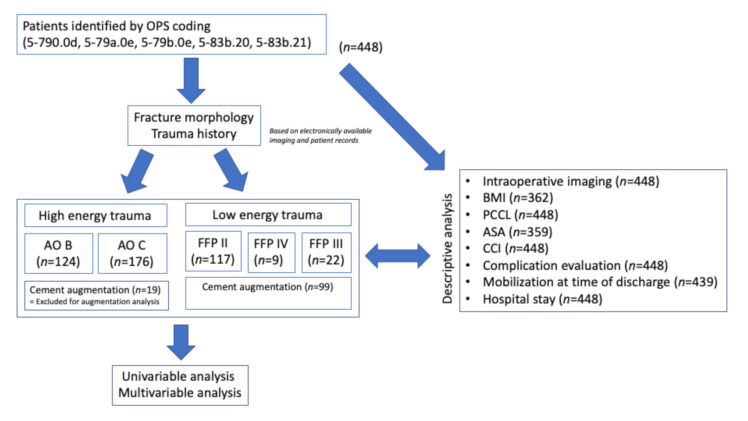
Schematic overview of the included cases according to the STrengthening the Reporting of OBservational studies in Epidemiology (STROBE) recommendations [32]. High-energy trauma was excluded from the analysis of augmentation. No follow-up was performed. Reduced numbers represent missing data in available sources. In all cases, a clinical report and a distinct radiological report or imaging was available. In cases of available electronic imaging, classification was performed by the authors. Patients without a history of high-energy trauma were re-classified using the fragility fractures of the pelvis (FFP) classification (FFP II—non-displaced posterior ring fracture; FFP III—displaced unilateral posterior pelvic ring fracture; and FFP IV—displaced bilateral posterior pelvic ring fracture). OPS = operation and procedure coding; AO = Arbeitsgemeinschaft für Osteosynthesefragen; FFP = Fragility fractures of the pelvis; BMI = Body mass index; PCCL = Patient Clinical Complexity Level; ASA = American Society of Anesthesiologists score; CCI = Charlson Comorbidity Index.

**Figure 2 jcm-09-02660-f002:**
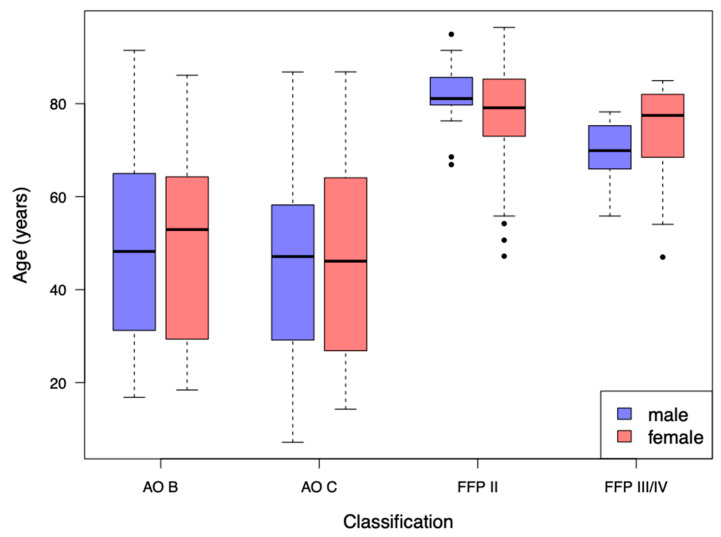
Descriptive analysis of age separated by sex in the different classification patterns. Dots indicating outliers.

**Figure 3 jcm-09-02660-f003:**
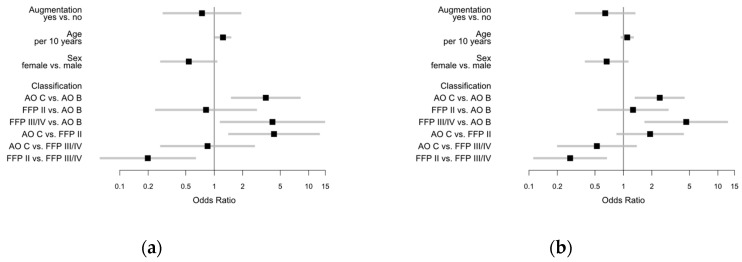
Odds ratios with 95% Wald confidence limits for (**a**) major complications (augmented vs. non-augmented 0.742; female vs. male: 0.538; AO C vs. AO B: 3.516; FFP II vs. AO B: 0.819; FFP III/IV vs. AO B: 4.137; AO C vs. FFP II: 4.293; AO C vs. FFP III/IV: 0.850; FFP II vs. FFP III/IV: 0.198) and (**b**) minor complications (augmented vs. non-augmented 0.642; female vs. male: 0.665; AO C vs. AO B: 2.423; FFP II vs. AO B: 1.263; FFP III/IV vs. AO B: 4.624; AO C vs. FFP II: 1.918; AO C vs. FFP III/IV: 0.524; FFP II vs. FFP III/IV: 0.273).

**Figure 4 jcm-09-02660-f004:**
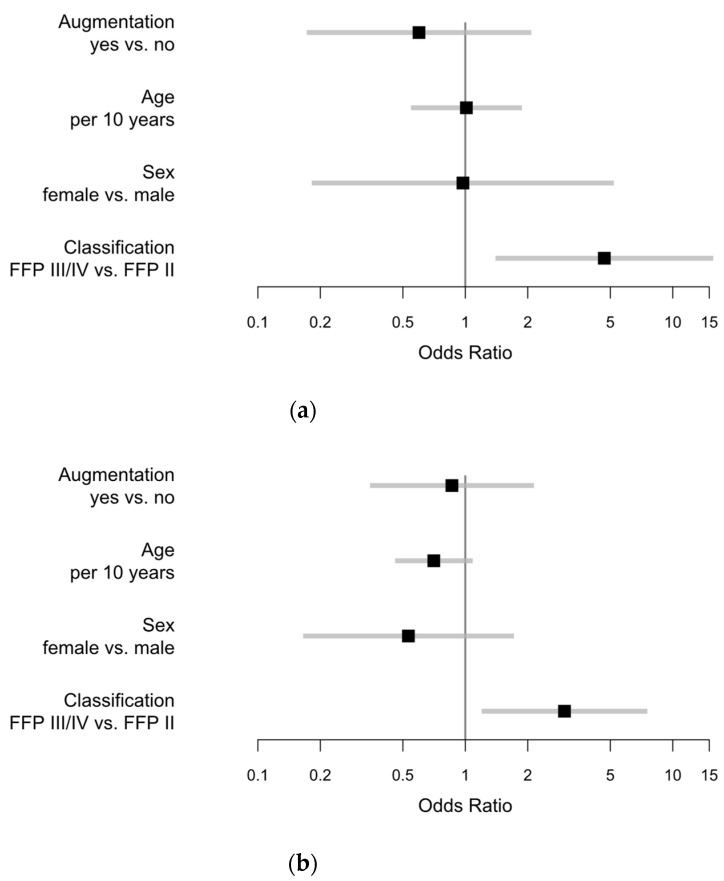
Odds ratios with 95% Wald confidence limits for (**a**) major complications (augmented vs. non-augmented: 0.598; female vs. male: 0.974; FFP III–IV vs. FFP II: 4.679) and (**b**) minor complications (augmented vs. non-augmented: OR 0.862; female vs. male: OR 0.532; FFP III–IV vs. FFP II: OR 3.006).

**Figure 5 jcm-09-02660-f005:**
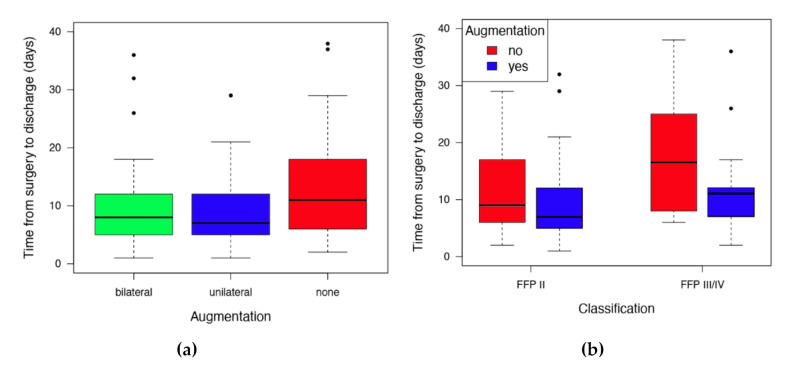
Postoperative duration of stay in the FFP cases: (**a**) bi-lateral and unilateral showing about a 25% shorter stay in hospital compared to the non-augmented group; (**b**) differentiation between the FFP II and FFP III/IV also shows the effect of augmentation with earlier discharge after augmentation. Dots indicating outliers.

**Figure 6 jcm-09-02660-f006:**
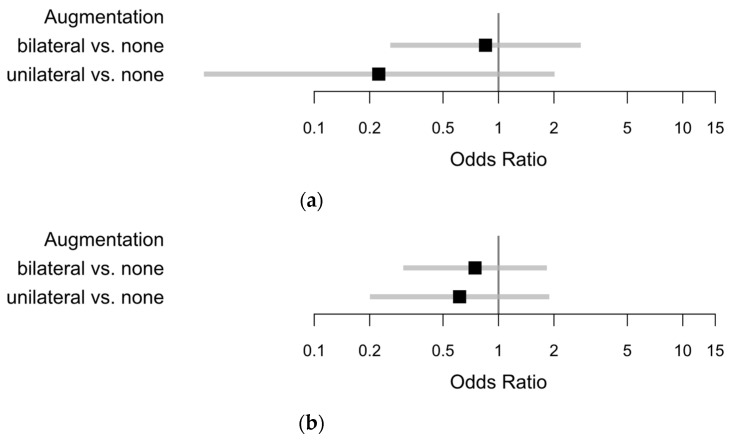
Univariable analysis of augmentation concerning (**a**) major complications; and (**b**) minor complications.

**Table 1 jcm-09-02660-t001:** Minor and major complications.

Minor Compilation	Major Complication
• Hypertensive derailment	• Inguinal hernia incarcerated postoperatively
• Delayed wound healing	• Cerebral hemorrhage postoperatively
• Infections (UTIs ^1^, pneumonia, other infections without severe clinical impact)	• Postoperative fracture dislocation
	• Intraoperative resuscitation
• Postoperative atrial fibrillation	• Death
• Postoperative anemia	• Implant failure requiring revision surgery
• Pain described to be the reason of delayed discharge	• Heavy intraoperative bleeding ^2^
• Superficial wound infections	• Acute kidney failure (AKIN)
• Hematoma (without revision)	• Persistent postoperative neurological impairment until discharge ^3^
• Temporary neurological impairment ^1^	• Postoperative cerebral insult
• Postoperative angioedema	• Wound infection requires revision surgery
• New diagnosis of restless leg syndrome	• Postoperative pulmonary artery embolism
• Early loosening of external supraacetabular fixator	• Postoperative wound-related hemorrhage
• Pain beyond a period of about 6 months	• Postoperative severe hematoma with revision surgery
	• Postoperative positive Trendelenburg sign

^1^ Urinary tract infection; ^2^ intraoperative transfusion or description of severe bleeding in the surgeon’s report; ^3^ Postoperative peripheral neurological impairment is reported separately.

**Table 2 jcm-09-02660-t002:** Mobilization at time of discharge.

	Free Mobilization	Rollator/Walking Frame	Wheelchair	Crutches/Walking Stick	Bedridden	No Information
Median age (years)	86.17	80.16	75.45	76.53	84.72	–
Q 25%	56.48	75.46	66.48	64.81	69.01	–
Q 75%	87.80	84.14	84.95	83.96	85.65	–
FFP II	2 (1.71%)	60 (51.28%)	7 (5.98%)	37 (31.62%)	9 (7.69%)	2 (1.71%)
FFP III	–	1 (11.11%)	1 (11.11%)	6 (66.67%)	–	1 (11.11%)
FFP IV	1 (4.55%)	16 (72.73%)	3 (13.64%)	1 (4.55%)	1 (4.55%)	–

**Table 3 jcm-09-02660-t003:** Duration of stay (number of days) after surgery and the occurrence of major complications in dependence of PCCL.

	PCCL					
	0–1	2	3	4	5–6	Total
*n*	33	18	50	37	10	148
Median (d)	9	5.5	8	9	25	8
Q 25%	5	4	5	6	16	5
Q 75%	12	8	11.5	15	32	13.5

PCCL: Patient Clinical Complexity Level.

**Table 4 jcm-09-02660-t004:** Mobilization at time of discharge in dependence of augmentation.

	Free Mobilization	Crutches/Walking Stick	Rollator/Walking Frame	Wheelchair	Bedridden	No Information
Unilateral augmentation	2 (6.06%)	10 (30.30%)	18 (54.55%)	1 (3.03%)	2 (6.06%)	–
Bilateral augmentation	–	12 (18.18%)	39 (59.09%)	7 (10.61%)	6 (9.09%)	2 (3.03%)
No augmentation	1 (2.04%)	22 (44.9%)	20 (40.82%)	3 (6.12%)	2 (4.08%)	1 (2.04%)
		In 3 cases, no information about mobilization could be found.				

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
