# Peer review of "Safety, Effect and Feasibility of Percutaneous SI-Screw with and without Augmentation—A 15-Year Retrospective Analysis on over 640 Screws"

_jcm, 2020, doi:10.3390/jcm9082660_

Round 1
Reviewer 1 Report
Dear Author
It's a njice paper with a long follow up
I suggest
to cite and discuss these paper:
For emergency
Falzarano G, Medici A, Carta S, Grubor P, Fortina M, Meccariello L, Ferrata P. The orthopedic damage control in pelvic ring fractures: when and why-a multicenter experience of 10 years' treatment. Journal of Acute Disease (2014):201-206. doi: 10.1016/S2221-6189(14)60044-5
For heteropic calcification:
Rollo G, Pellegrino M, Filipponi M, Falzarano G, Medici A, Meccariello L, Bisaccia M, Piscitelli L, Caraffa A. A case of the management of Heterotopic ossification as the result of acetabular fracture in a patient with traumatic brain injury. International Journal of Surgery Open 1 (2015) 30–34. DOI:10.1016/j.ijso.2016.03.001
Reviewer 2 Report
The paper is well written. I think the paper is worth being published in order to
1)Allow the surgeon to better inform the patient preoperatively.
2)Repeat and demonstrate the anatomical structures postoperatively with the patient in the follow-up examination
Moreover, the authors' methodology is very scientific and systematic.
I think this paper with this information is worth publishing.
Reviewer 3 Report
Very interesting paper based on large and thoroughly analysed clinical data.
Well documented conclusions.
The only shortcomings concern the presentation of the results, which could be shown graphically to a greater extent.
Fig. 1 and Fig. 4b should be corrected, where it is not possible to distinguish the presented results (women / men; augmentation no / yes)

Reviewer 4 Report
Overall, this is very interesting data that undoubtedly adds to the body of evidence on the topic of SI screw application and the optional screw augmentation. Unfortunately the way the data is presented will impair its significance to the scientific community and the general readership.
Please consider the following requests and counsels to improve the paper:
Although the authors argue for the benefit of including both high and low energy pelvic fractures, I strongly believe that this is the major factor to reduce the paper's readability and to dilute its scientific value. There simply is too much data and groups are too heterogeneous, weakening the report in regards to perspicuity and validity. In my opinion, the authors should concentrate on safety and feasibility of either SI screw placement in general or SI screw augmentation. Two independent studies would be possible as well, even if this means, of course, that it will result in smaller sample sizes.
Either way, the study should be written in concordance with the STOBE guidelines. A flow chart in line with these guidelines should definitely be included to describe the patient cohort, inclusion and exclusion criteria etc. For example, it is not clear to me, if all patients actually received a pelvic CT scan postoperatively, which I would assume to be an important inclusion criterion. Same applies for FFP preoperatively, since a pelvic CT is indispensable to establish the proper diagnosis of FFP.
In regards of complications:
- please define exactly implant/cement related complications
- I assume the authors refer to peripheral and not central neurologic impairment, please indicate correctly. Also 'neurogenic' seems to be misplaced in this context.
- define 'heavy intraoperative bleeding'
- is the threshold of screw misplacement 3.5mm or 3.6mm, please specify?
The present study does not specifically comment on concomitant anterior pelvic ring fixation. Is data on this available? May it have a role in complication rates? Please discuss.
line 36-38
Does the given reference effectively support this sentence? What about percutaneous ilio-lumbal fixation or percutaneous posterior pelvic ring plate osteosynthesis?
Please consider to streamline the discussion section in regards of redundancy.
Round 2
Reviewer 4 Report
Overall, the paper improved with the recent changes. Thank you for considering the recommended modifications.